# CLTA-4 Expression Is Associated with the Maintenance of Chronic Inflammation in Endometriosis and Infertility

**DOI:** 10.3390/cells10030487

**Published:** 2021-02-25

**Authors:** Monika Abramiuk, Dominika Bębnowska, Rafał Hrynkiewicz, Paulina Niedźwiedzka-Rystwej, Grzegorz Polak, Jan Kotarski, Jacek Roliński, Ewelina Grywalska

**Affiliations:** 1Department of Gynecological Oncology and Gynecology, Medical University of Lublin, 20-081 Lublin, Poland; monika.abramiuk@gmail.com (M.A.); grzegorz.polak@umlub.pl (G.P.); jan.kotarski.gabinet@gmail.com (J.K.); 2Institute of Biology, University of Szczecin, 71-412 Szczecin, Poland; dominika.bebnowska@usz.edu.pl (D.B.); rafal.hrynkiewicz@usz.edu.pl (R.H.); 3Department of Clinical Immunology and Immunotherapy, Medical University of Lublin, 20-093 Lublin, Poland; jacek.rolinski@gmail.com (J.R.); ewelina.grywalska@gmail.com (E.G.)

**Keywords:** CTLA-4, endometriosis, immune response, infertility, NK cells, NKT-like cells, negative co-stimulation

## Abstract

Altered immune mechanisms are implicated in the pathogenesis of endometriosis. CTLA-4 is a membrane receptor that favors the anergic state of lymphocytes, which may disrupt the immune system response in the endometriotic environment. In this study, we examined the expression of CTLA-4 on T and B cells by flow cytometry and its levels in blood serum and peritoneal fluid by ELISA. Levels of CTLA-4+ T cells were significantly higher in patients with more advanced endometriosis than in those with less advanced disease. Additionally, the negative correlation of CTLA-4^+^ T lymphocytes and the percentage of NK and NKT-like cells in women with endometriosis and infertility may indicate a different etiopathogenesis of endometriosis accompanying infertility. Our findings shed light on the potential of CTLA-4 in developing new diagnostic and therapeutic approaches in endometriosis management.

## 1. Introduction

Endometriosis belongs to the group of gynecological diseases with a mild course. It is a chronic disease, often recurrent, consisting of the pathological presence of endometrial tissue outside its proper place of occurrence, that is, the uterine cavity. The growth of endometrial tissue is closely associated with steroid metabolism; therefore, it is estimated that the disease affects approximately 10% of all women of reproductive age [1,2,3]. Incidence rates vary considerably depending on the population: from 5 to 50% of women treated for infertility to 5 to 21% of women with pelvic pain syndrome [2,4,5,6,7]. This means that around 176 million women are affected by endometriosis worldwide [8]. The diverse presentation of the disease-from superficial changes in the pelvic peritoneum to deeply infiltrating changes involving the intestines-is associated with a wide range of accompanying symptoms that can be divided into two groups: pain and infertility [9,10]. Although chronic inflammation and estrogen dependency are well-established factors in the characteristics of endometriosis, the exact etiology of the disease remains largely elusive [11,12]. This can be attributed to the complex and multifactorial nature of the disease in which specific genetic, hormonal, environmental and immunological factors have been identified [13].

Recently, much attention has been paid to altered mechanisms of the immune response. It is still unclear how immune system dysfunction affects the pathogenesis of endometriosis and whether it is involved in the onset of the disease. The classic theories of the development of endometriosis include the theory of Dmowski [14], assuming that the survival and development of endometrial tissue outside the uterine cavity is conditioned by the presence of local immunological tolerance in its environment. As a result of creating specific conditions of immune deficit, it is impossible to effectively eliminate from the peritoneal cavity the morphotic elements that got into it as a result of retrograde menstruation. The expression of a defect in the immune response that allows the creation of such conditions was to be, among others, altered percentage and activity of selected cell subpopulations: NK cells (natural killer cells), cytotoxic lymphocytes (CTL) and macrophages. Other reports suggest that fragments of the endometrium of patients with endometriosis have an increased adhesive capacity [15]. Additionally, the eutopic endometrium is characterized by an increased concentration of metalloproteinases and plasminogen activators, which indicates the possibility of increased proteolysis of endometrial fragments entering the peritoneal cavity through retrograde menstruation [16,17]. These fragments, when implanted into the peritoneal cavity, trigger a response from the innate and adaptive components of the immune system. The concentration of immune system cells around the lesions increases and the mechanisms related to tissue repair are activated. However, the inability to counteract the permanent presence of endometrial fragments may, as a consequence, lead to “overloading of the immune system” and then to its proper functioning [18].

The CTLA-4 molecule belongs to the family of type I membrane receptors and is an important checkpoint in signaling between cells of the immune system. Its action is inhibitory and is a signal of negative feedback in the development of a specific immune response [19]. CTLA-4 expression is observed both on the surface of CD4+ T, regulatory T lymphocytes (Treg), as well as on the B 19+ cells subjected to antigenic stimulation. Stimulated T cells express CTLA-4 in the presence of CD28. Both molecules have the ability to interact with the same ligands – B.7-1 (CD80) or B.7-2 (CD86) but CTLA-4, according to various sources, has a 10–50 times greater affinity for them [20,21]. CTLA-4 promotes anergy state of lymphocytes. At the cellular level, the synthesis of cyclin D3 and cdk4/cdk6 kinases is inhibited, degradation of the p27 inhibitor protein and expression of cyclin D2 are enhanced. Even after recognition of a specific antigen, cells are not activated, resulting in no transition from G0 to G1. The costimulatory signals that induce the expression of the virgin lymphocyte are not sufficient in this case [22,23,24,25].

The aim of the study is to evaluate the expression of CTLA-4 on the surface of T and B lymphocytes and to determine the concentration of soluble CTLA-4 antigen in peripheral blood plasma and peritoneal fluid in patients as well as to analyze the relationship between these data and clinical parameters in patients with endometriosis.

## 2. Materials and Methods

### 2.1. Patiens and Controls

The study included 74 women hospitalized and diagnosed at the 1st Department of Oncological Gynecology and Gynecology of the Medical University of Lublin in the years 2016–2018. The study group consisted of 54 women with newly diagnosed endometriosis confirmed by histopathological examination. The stage of disease was assessed according to the revised American Society for Reproductive Medicine (rASRM) scale. The control group consisted of 20 patients without any signs of endometriosis during laparoscopy. The study and control groups were created taking into account the inclusion criteria, which included the lack of infertility treatment (possible prior diagnostics in this area) and the lack of taking medications affecting the immune system. Patients enrolled in the study did not show any signs of infection in the two months preceding the study, and did not have any blood transfusions. The study and control groups were selected from patients without autoimmune diseases, allergies or cancers. The study protocol was assessed by the Bioethics Committee of the Medical University of Lublin and received a positive opinion with the number KE-0254/302/2014. All the patients got to know the subject of the study and gave informed consent to participate in it.

### 2.2. Material Collection

The material for the research was peripheral blood (PB) taken from the cephalic vein or other superficial vein with the best access for phlebotomy at cubital fossa from the patients in a total amount of 15 mL. Blood was drawn the day before the planned surgery. To obtain serum for measuring CTLA-4 concentration, 5 mL of PB were collected using clot activator tubes. Then 10 mL was taken into an ethylene diamine tetraacetate (EDTA; Sarstedt aspiration vacuum systems, Germany) tube for isolation of PB mononuclear cells (PBMC) and flow cytometric analysis. In patients with endometriosis, an additional 5 mL of peritoneal fluid was collected in EDTA tubes. The peritoneal fluid was collected immediately after insertion of the laparoscope to avoid blood contamination.

### 2.3. Immunophenotyping

PB samples collected in EDTA tubes were diluted 1:1 with 0.9% phosphate-buffered saline (PBS) magnesium (Mg^2+^) and calcium (Ca^2+^) ions-free (Biochrome AG, Berlin, Germany). The blood in the obtained dilution was layered on 3 mL of Gradisol L (1.077 g/mL) (Aqua Medica, Poland) and then centrifuged for 20 min in a density gradient (700× *g*). The obtained fraction of peripheral blood mononuclear cells (PBMC) was collected with Pasteur pipettes and washed twice in PBS by vortexing for 5 min. Then, using a Neubauer chamber, the number of cells was determined. Viability was determined using trypan blue (0.4% Trypan Blue Solution, Sigma Aldrich, St. Louis, MO, USA).

Flow cytometry was used to assess the percentages of CTLA-4+ cells within the CD4+ T, CD8+ T, and CD19+ B lymphocyte populations. The analysis was performed using a FACSCalibur flow cytometer (Becton Dickinson, Franklin Lakes, NJ, USA). The acquisition and analysis of the results were performed using the CellQuest program (Becton Dickinson, Franklin Lakes, NJ, USA). A CaliBRITE calibration kit (Becton Dickinson, Franklin Lakes, NJ, USA) was used to optimize the flow cytometer settings.

After the PBMCs were isolated, the cell suspension was divided into single tubes with 1 × 10^6^ cells per sample and incubated with the appropriate monoclonal antibodies (mAbs). We used fluorochrome-conjugated mAbs against the following markers: CD45− fluorescein isothiocyanate (FITC)/CD14− phycoerythrin (PE), mouse anti-human CD3-CyChrome, mouse anti-human CD19-FITC, mouse anti-human CD4-FITC, mouse anti-human CD8-FITC, mouse anti-human CTLA-4-PE-Cyanine 5 (Cy5) (BD Biosciences, San Jose, CA, USA). We also used the Human Treg Flow kit (FOXP3 Alexa Fluor 488/CD4 PE-Cy5 Cy5/CD25 PE; BioLegend, San Diego, CA, USA) to identify the CD4+CD25+high forkhead box P3 (FOXP3+) Treg subpopulation. During analysis, the CD3+CD16+CD56+ natural killer T-like (NKT-like) cells population and CD16+CD56+ NK+ cells were also measured with anti-CD3-FITC, CD16CD56-PE, and CD45-peridinin–chlorophyll–protein (PerCP) mAbs (BD Biosciences, San Jose, CA, USA). The cells were incubated for 20 min at room temperature with 20 μL of each mAb per sample. Next, the suspension was washed twice with PBS (700× *g*, 5 min) and analyzed in a flow cytometer equipped with an argon laser (λ = 488 nm) that operated at 20,000 cells per run. The labeled cells were examined based on lymphocyte gates at combined CD45/CD14 coordinates. The samples were gated on forward scatter vs. side scatter. The results of the flow cytometry analysis are presented as percentage of stained cells (Figure 1).

### 2.4. CTLA-4 Concentration Measurement

The determination of CTLA-4 antigen in peripheral blood plasma and peritoneal fluid supernatant was performed using the enzyme-linked immunosorbent assay (ELISA) method using CTLA-4 (Soluble) Human ELISA Kit with a sensitivity of 0.13 ng/mL (ThermoFisher Scientific, Waltham, MA, USA). The test procedure was performed in accordance with the recommendations of manufacturer. The results were read by using the VICTOR3 automatic reader (Perkin Elmer, Boston, MA, USA). It measures the light absorbance in the test material and compares it with a known concentration control sample. The WorkOut computer program, cooperating with the reader, on the basis of known concentrations, drew linear curves on the basis of which the concentration of soluble antigens and cytokines in the tested samples was calculated.

### 2.5. Statistical Analysis

The obtained results were analyzed statistically. Results from measurable parameters are presented as the mean, median, minimum, and maximum values and standard deviation. Nonmeasurable parameters are presented as means of count and percentage. The normality of the distribution of variables in the studied groups was checked using the Shapiro–Wilk normality test. The Student’s t-test was used to test the differences between the two groups, and in the case of non-fulfillment of the conditions for its use-the Mann–Whitney U test. The comparison of three or more groups was performed using ANOVA with Tukey’s RIR test, and in the case of failure to meet the conditions for its application the Kruskal–Wallis test with multiple comparisons of mean ranks for all trials with Bonferroni’s correction was used. The relationships between the variables were assessed on the basis of the Pearson’s R correlation coefficient, and if outliers were detected, the Spearman’s rank correlation was used. A significance level of *p* < 0.05 was adopted, indicating the existence of statistically significant differences or relationships. The database and statistical surveys were prepared and carried out using the Statistica computer software, ver. 13.3 (StatSoft, Kraków, Poland).

## 3. Results

### 3.1. Patiens and Control Group

The study group consisted of 54 women with newly diagnosed endometriosis confirmed by histopathological examination. The endometriotic lesions were characterized by endometrial cysts and changes on the peritoneal surface of Douglas sinus, sacro-uterine ligaments or parietal abdominal peritoneum.

The mean ± standard deviation (SD) age of the patients was 35 ± 6.2 years (median, 35 years; range, 22–48 years). According to the rASRM scale, 17 (31%) patients were classified as Stage I, 17 (31%) as Stage II, 9 (17%) as Stage III and 11 (20%) as Stage IV.

Infertility was found in 27 women (50%), peritoneal adhesions were found during surgery in 31 (57%), and 43 (80%) reported the presence of pelvic pain syndrome. Eleven patients had deep infiltrating endometriosis, and among 31 who had peritoneal adhesions, 20 (70%) were diagnosed with endometrioma.

The control group consisted of 20 patients without any signs of endometriosis during laparoscopy. The reason for the performed procedures was in all cases the control of the patency of the fallopian tubes, without any irregularities in this regard. The mean ± SD age of the patients was 37 ± 9 years (median, 36 years; range, 26–53 years).

### 3.2. Expression of the CTLA-4 Molecule on CD4+ T Cells, CD8+ T Cells and CD19+ B Cells in Patients with Endometriosis and the Control Group

Table 1 shows the percentage of CD4+ T cells, CD8+ T cells and CD19+ B cells expressing the CTLA-4 antigen among peripheral blood lymphocytes in patients with endometriosis and in the control group.

Patients with endometriosis had significantly higher percentage of CD8+CTLA-4+ T lymphocytes expressing the CTLA-4 antigen in the peripheral blood than individuals from the control group (*p* < 0.001, Table 1).

Percentages of T CD4+CTLA-4+ and B CD19+CTLA-4+ lymphocytes were not significantly different between patients with endometriosis and the control group (Table 1).

Patients with endometriosis were divided into three groups depending on the stage of rASRM. The group 1 (Stage I) and group 2 (Stage II) were represented by 17 patients each, and the group 3 (Stage III-IV) by 20 individuals. A positive correlation between the severity of endometriosis and the percentage of CD4+CTLA-4+ T cells (Spearman’s R = 0.531, *p* < 0.001), and a positive correlation between the severity of endometriosis and the percentage of CD8+CTLA-4+ T cells (Spearman’s R = 0.450, *p* = 0.001) were observed. The severity of endometriosis was weakly correlated with the percentage of CD19+CTLA-4+ B cells (Spearman’s R = 0.315, *p* = 0.02).

In the subgroup of patients with endometriosis and accompanying infertility or pelvic pain syndrome, no statistically significant differences were observed in the percentages of lymphocytes expressing the CTLA-4 antigen. The percentages of CD4+CTLA-4+ and CD8+CTLA-4+ T lymphocytes were significantly higher in women with endometriosis and coexisting adhesion disease than in women with endometriosis but no adhesion disease (Table 2).

#### 3.2.1. Percentage of T and B Lymphocytes Expressing CTLA-4 and Parameters of Specific and non-specific responses in patients with endometriosis and accompanying infertility

A positive correlation of the percentage of CD4+CTLA-4+ T cells with the number of monocytes was observed) (Figure 2a). However, in the case of CD4+CTLA-4+ T cells with the percentage of CD3-CD16+CD56+ NK cells (Figure 2b), and the percentage of NKT-like CD3+CD16+CD56+ cells, negative correlations were shown (Figure 2c).

A positive correlation between the percentage of CD8+CTLA-4+ T cells and the number of monocytes was observed (Figure 3a). There was a negative correlation of CD8+CTLA-4+ T cells with the percentage of CD3-CD16+CD56+ NK cells (Figure 3b) and the percentage of NKT-like CD3+CD16+CD56+ cells (Figure 3c). Moreover, a positive correlation was found between the percentage of CD8+CTLA-4+ T cells and the percentage of CD4+CD25+highFoxp3 regulatory T cells (Figure 3d).

A negative correlation between CD19+CTLA-4+ B lymphocytes and the percentage of CD3-CD16+CD56+ NK cells was reported (Figure 4).

#### 3.2.2. Percentage of T and B Lymphocytes Expressing CTLA-4 and Parameters of the Specific and Non-Specific Response in Patients with Endometriosis and Accompanying Adhesive Disease

In patients with endometriosis and coexisting adhesive disease, negative correlation of T lymphocytes CD4+CTLA-4+ with the percentage of NK cells CD3-CD16+CD56+ was demonstrated (R = −0.622, *p* < 0.001). The percentage of T cells CD4+CTLA-4+ was negatively correlated with the percentage of Treg cells CD4+CD25+highFoxp3+ (R = −0.45, *p* = 0.013).

In addition, the percentage of CD8+CTLA-4+ T cells was positively correlated with the number of monocytes (R = 0.396, *p* = 0.028), whereas the percentage of CD8+CTLA-4+ T cells was negatively correlated with the percentage of CD3-CD16+CD56+ NK cells (R = −0.522, *p* = 0.003) and with the percentage of CD3+CD16+CD56+ NKT-like cells (R = −0.373, *p* = 0.039).

#### 3.2.3. Percentage of T and B Lymphocytes Expressing CTLA-4 and Selected Parameters of the Specific and Non-Specific Response in Patients with Endometriosis and Accompanying Pelvic Pain Syndrome

In patients with endometriosis and pelvic pain syndrome, we observed a positive correlation of the percentage of CD4+CTLA-4+ T lymphocytes to the number of monocytes (R = 0.381, *p* = 0.013), and negative correlations of CD4+CTLA-4+ T lymphocytes to the percentage of CD3-CD16+CD56+ NK cells (R = −0.599, *p* < 0.001) and the percentage of CD3+CD16+CD56+ NKT-like cells (R = −0.364 *p* = 0.018). There was also a positive correlation between the percentage of CD4+CTLA-4+ T cells and the percentage of CD4+CD25+highFoxp3 Treg cells (R = 0.315, *p* = 0.042).

In addition, a positive correlation of the CD8+CTLA-4+ T lymphocyte percentage to the monocyte count (R= 0.464, *p* = 0.02) and negative correlations of CD8+CTLA-4+ T lymphocytes to CD3-CD16+CD56+ NK cell percentage (R = −0, 497, *p* = 0.001) and the percentage of NKT-like CD3+CD16+CD56+ cells (R = −0.475, *p* = 0.001) were reported.

In this subgroup of patients, a negative correlation of CD19+CTLA-4+ B lymphocytes with the percentage of CD3-CD16+CD56+ NK cells (R = −0.493, *p* = 0.001) and a positive correlation between the percentage of CD19+CTLA-4+ B lymphocytes and the percentage of CD4+CD25+highFoxp3 Treg cells (R = 0.419, *p* = 0.006) were also observed.

### 3.3. Soluble CTLA-4 Antigen Concentration in Peripheral Blood Plasma and Peritoneal Fluid

The concentration of soluble CTLA-4 antigen in the peripheral blood plasma was significantly higher in women with endometriosis (mean, 7.12; SD, 1.55) than in the control group (mean, 6.13; SD, 1.52; *p* = 0.01).

No statistically significant differences between the severity of endometriosis and the concentration of soluble CTLA-4 antigen in the peripheral blood plasma in patients with endometriosis were observed. Moreover, no statistically significant differences were found in the soluble CTLA-4 antigen concentration in the peripheral blood plasma in patients with endometriosis and coexistence of infertility, pelvic pain syndrome or adhesion disease.

The concentration of CTLA-4 antigen in the peritoneal fluid in patients with endometriosis ranged from 0.19 to 7.56, and the mean (SD) was 2.56 (1.90) ng/mL.

No statistically significant differences were found between the severity of endometriosis and the concentration of soluble CTLA-4 antigen in the peritoneal fluid in patients with endometriosis.

No statistically significant differences in the concentrations of soluble CTLA-4 antigen in the peritoneal fluid between the subgroups of patients with endometriosis and coexistence of infertility, pelvic pain syndrome, or adhesion disease were reported.

#### Relationship between the Concentration of Soluble CTLA-4 Antigen and Other Parameters of Specific and Non-Specific Responses in Patients with Endometriosis and with Accompanying Infertility and Pelvic Pain Syndrome

A positive correlation between the concentration of soluble CTLA-4 antigen in the peritoneal fluid and the percentage of Treg lymphocytes CD4+CD25+highFoxp3 (R = 0.42, *p* = 0.029) was observed.

In patients with endometriosis and pelvic pain syndrome, a weak negative correlation was observed between the concentration of soluble CTLA-4 antigen in peripheral blood plasma and the percentage of CD4+CD25+highFoxp3 Treg cells (R = 0.31, *p* = 0.043).

## 4. Discussion

The mechanism of regulating the immune response takes place in two stages. After initial antigen-mediated activation of the lymphocyte, a costimulatory signal occurs along with the major histocompatibility complex (MHC) and T-cell receptor (TCR). This can modify the further course of the process. The activation signal from the interaction between CD28 on T cells and CD80 (B7.1) and CD86 (B7.2) on antigen presenting cells (APC) leads to lymphocyte proliferation, increased survival and differentiation through cytokine production. CTLA-4 is one of the first negative regulators to compete directly with CD28 by binding to the same ligands. Strong activation of TCR results in increased expression of CTLA-4 on the surface of the lymphocyte. As a result, after binding to CD80/CD86, the lymphocyte is put into anergic state, characterized by inhibited IL-2 production and further proliferation. The CTLA-4 molecule is also involved in other aspects of regulating the immune response [26]. In animal models, genetic CTLA-4 deficiency attenuated Treg suppressor functions. Constitutive expression of CTLA-4 on Treg can sequester or internalize the CD80/CD86 receptor on APC cells, leading to a reduction in CD28 costimulation. The lack of a costimulatory signal can lead to reduced proliferation of T cells and subsequent reduction in their effector functions. CTLA-4 blockade directly inhibits the autoimmune response in vitro in a mouse model of endometriosis. By using an anti-CTLA-4 antibody, a gradual reduction of CD4+CD25+ Treg cells was demonstrated. In addition, through broken immunotolerance, inhibited proliferation and invasion of ectopic endometrial cells was observed [26].

Studies on gene polymorphism for the CTLA-4 molecule did not show a statistically significant relationship with the incidence of endometriosis [27,28]. However, this does not mean that the molecule is not involved in the local disturbance of the immune response in the disease environment. Experiments on the therapeutic potential of the anti-CTLA-4 antibody in a mouse model have shown that it may be an important tool in inhibiting the progression of endometriosis by regulating the overproduction of CD4+CD25+ Treg cells, a phenomenon repeatedly described in the literature [26].

In our study, a significantly higher expression of the CTLA-4 antigen was observed among CD8+T cells in the peripheral blood in patients with endometriosis than in the control group. In CD4+ T and CD19+CTLA-4+ B cells, no such differences were observed between the group of patients with endometriosis and the control group. At the same time, a positive correlation was demonstrated between the stage of endometriosis and the percentage of CD4+CTLA-4+ T cells and CD8+CTLA-4 T cells. There was also a weak positive correlation between the stage of endometriosis and the percentage of CD19+CTLA-4+ B lymphocytes. These data seem interesting in the context of the reports of Hegel et al. [29], who proved that CD8+ T cells without CTLA-4 expression show significantly higher IFN-γ and granzyme B production as well as enhanced cytolysis in an animal model. At the same time, CTLA-4 expression had no effect on the proliferation of these cells [29]. Recently, patients with endometriosis were reported to have higher percentage of Treg cells expressing CTLA-4 in plasma than healthy individuals [30]. No such relationship was found in Treg cells isolated from peritoneal fluid [30].

In the group of patients with endometriosis associated with infertility, a negative correlation was found between the number of CD4+CTLA-4 lymphocytes and the percentage of NK cells and the percentage of NKT-like CD3+CD16+CD56+ cells. A similar, negative correlation was found when examining CD8+ lymphocytes. A positive correlation was found between the number of CD8+CTLA-4 lymphocytes and the percentage of Treg CD4+CD25+ highFoxp3. All these data additionally indicate the important role of immunosuppressive mechanisms in the development of the disease. Although the clinical relationship between endometriosis and infertility is evident, the causal link between endometriosis and infertility is still poorly established [31]. Anupa et al. [32] in their recent study showed that the secretory profile of inflammation-associated cytokines in a eutopic endometrium derived from patients suffering from severe ovarian endometriosis during their “window of implantation” differs from those of other infertile patients [32]. CCL3, CCL4, CCL5, CXCL10, FGF2, IFNG, IL1RN, IL5, TNFA, and VEGF were detected exclusively in stromal cells isolated from patients with endometriosis, and therefore might provide useful tool for distinguishing the source of infertility [32]. Our findings highlight the potential implication of CTLA-4 in the pathogenesis of endometriosis-related infertility and expression of CTLA-4 antigen on lymphocytes as a putative biomarker of this condition.

Moreover, in patients with endometriosis and intraoperative adhesions, significantly higher percentages of CD4+CTLA-4 and CD8+CTLA-4 T cells were observed. This information is somewhat in opposition to the reports by Holsti et al. [33], who, using a mouse model, investigated the effect of costimulatory signals on the formation of postoperative adhesions. According to their reports, blocking the interaction of CD28 with CD80 and CD86 ligands completely abolishes the formation of adhesions. However, the inhibitory costimulatory signal mediated by CTLA-4 does not significantly affect their formation [34].

A native, soluble form of CTLA-4 has been reported [35]. The presence of high serum concentrations of this form has been correlated with several autoimmune diseases [36]. Interestingly, Santoso et al. [37] revealed a significantly higher peritoneal concentration of soluble CTLA-4 in infertile endometriosis patients than in infertile patients without endometriosis. The differences in serum CLTA-4 levels were reported only between late-stage endometriosis and non-endometriosis patients [37]. Our study showed significant differences in plasma CTLA-4 concentration between patients with endometriosis and the control group, but no relationship between endometriosis stage and levels of CTLA-4 both in the peripheral blood and in the peritoneal fluid was observed.

## 5. Conclusions

Disturbed mechanisms of the immune response play an important role in the development of endometriosis. Statistically significant changes in the percentage of T lymphocytes expressing CTLA-4 antigen in the peripheral blood of women with endometriosis indicate a significant role of negative costimulation in the development of the disease and the persistence of chronic inflammation. It seems that a high percentage of anergic lymphocytes prevents the correct diagnosis and elimination of factors leading to excessive endometrial proliferation, as well as the removal of ectopic endometrial cells. Additionally, the negative correlation of CTLA-4+ T cells with the percentage of NK and NKT-like cells in women with endometriosis and infertility may indicate a different etiopathogenesis of endometriosis accompanying infertility. Moreover, the positive correlation of the CTLA-4 T-cell rate with the severity of endometriosis makes the CTLA-4 molecule valuable in developing new approaches to understanding and eliminating endometriosis, but more research is needed to confirm this potential.

## Figures and Tables

**Figure 1 cells-10-00487-f001:**
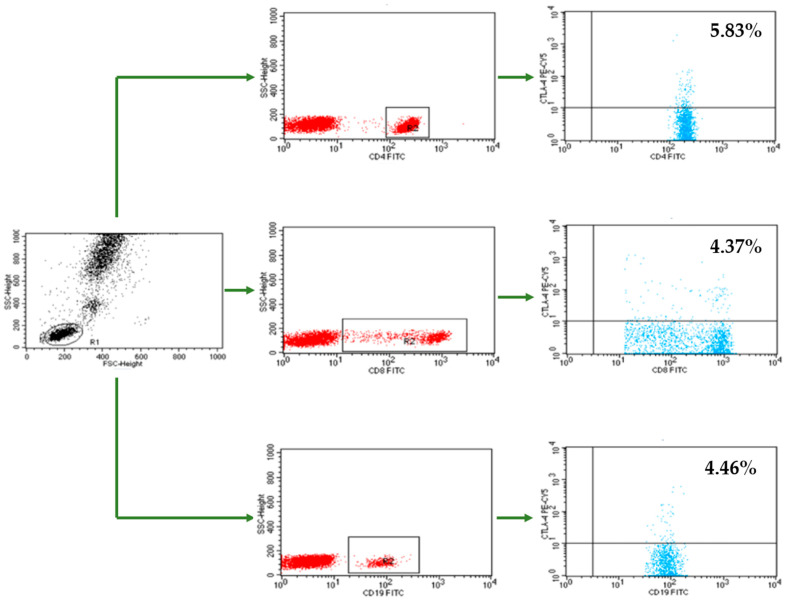
An example of flow cytometry analysis-assessment of the percentage of CD4+ T cells, CD8+ T cells and CD19+ B cells with CTLA-4 expression in patients with endometriosis. PBMCs from endometriosis patients were isolated and labeled as described in Immunophenotyping section. Scatter dot plots represent data from PBMCs in one endometriosis patient, analyzed by flow cytometry for CTLA-4 expression.

**Figure 2 cells-10-00487-f002:**
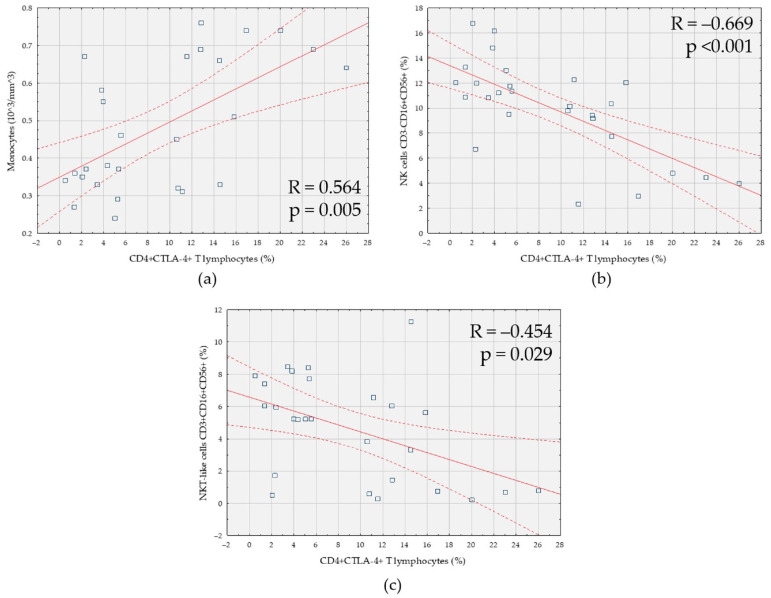
Correlation between the percentage of CD4+ T cells expressing CTLA-4 (CD4+CTLA-4+) and: (**a**) the number of monocytes; (**b**) the percentage of NK cells (CD3-CD16+CD56+); (**c**) the percentage of NKT-like cells (CD3+CD16+CD56+) in patients with endometriosis and coexisting infertility (*n* = 27).

**Figure 3 cells-10-00487-f003:**
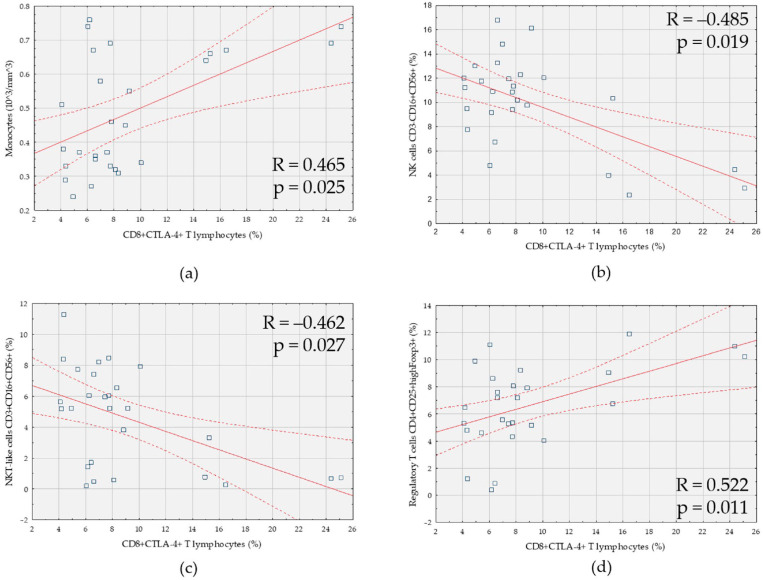
Correlation between the percentage of CD8+ T cells expressing CTLA-4 (CD8+CTLA-4+) and: (**a**) the number of monocytes; (**b**) the percentage of NK cells (CD3-CD16+CD56+); (**c**) the percentage of NKT-like (CD3+CD16+CD56+) cells; (**d**) the percentage of subpopulation of regulatory T cells with high forkhead box P3 expression (CD4+CD25+highFoxp3+) in patients with endometriosis and coexisting infertility (*n* = 27).

**Figure 4 cells-10-00487-f004:**
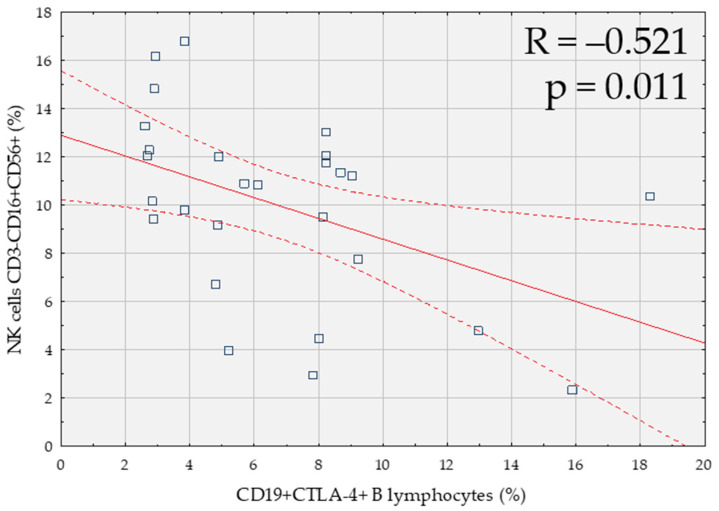
Correlation between the percentage of B lymphocytes expressing CTLA-4 (CD19+CTLA-4+) and the percentage of NK cells (CD3-CD16+CD56+) in patients with endometriosis and coexisting infertility (*n* = 27).

**Table 1 cells-10-00487-t001:** Percentage of CD4+ T lymphocytes, CD8+ T lymphocytes and CD19+ B lymphocytes expressing the CTLA-4 antigen among lymphocytes in peripheral blood in the group of patients with endometriosis and the control group.

Parameter	Group	Mean	Median	Min.	Max.	± SD	*p*-Value
Percentage of CD4+CTLA-4+ T lymphocytes (%)	EMS (*n* = 54)	9.22	10.10	0.11	30.05	6.66	NS
	Control (*n* = 20)	6.73	6.67	3.91	8.91	1.26	
Percentage of CD8+CTLA-4+ T lymphocytes (%)	EMS (*n* = 54)	8.90	7.20	4.00	25.11	4.69	<0.001
	Control (*n* = 20)	5.18	5.45	1.36	8.54	1.87	
Percentage of CD19+CTLA-4+ B lymphocytes (%)	EMS (*n* = 54)	6.83	5.72	2.56	18.28	3.93	NS
	Control (*n* = 20)	5.35	5.68	2.48	8.47	1.92	

EMS = endometriosis; ± SD = standard deviation; Min. = minimum; Max. = Maximum; NS = not significant.

**Table 2 cells-10-00487-t002:** Percentage of CD4+ T lymphocytes, CD8+ T lymphocytes and CD19+ B lymphocytes expressing the CTLA-4 antigen among peripheral blood lymphocytes in women with endometriosis depending on the coexisting adhesion disease.

Parameter	Group	Mean	Median	Min.	Max.	± SD	*p*-Value
Percentage of CD4+CTLA-4+ T lymphocytes (%)	Occurs	11.57	10.78	0.11	30.05	7.02	0.005
	Absent	6.05	5.04	0.53	15.51	4.63	
Percentage of CD8+CTLA-4+ T lymphocytes (%)	Occurs	10.50	7.91	4.07	25.11	5.50	0.005
	Absent	6.75	6.58	4.00	10.49	1.85	
Percentage of CD19+CTLA-4+ B lymphocytes (%)	Occurs	7.52	6.13	2.56	18.28	4.61	NS
	Absent	5.91	5.68	2.61	9.92	2.57	

SD = standard deviation; Min. = minimum; Max. = maximum; NS = not significant.

## Data Availability

Due to privacy and ethical concerns, the data that support the findings of this study are available on request from the corresponding author, [P.N.-R.].

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
