# Peer review of "CLTA-4 Expression Is Associated with the Maintenance of Chronic Inflammation in Endometriosis and Infertility"

_cells, 2021, doi:10.3390/cells10030487_

Round 1

Reviewer 1 Report

The manuscript “ CLTA-4 expression is associated with the maintenance of 2 chronic inflammation in endometriosis and infertility ” describes the expression of CTLA-4 on T and B cells, as well as the level of this marker in blood serum and peritoneal fluid, in women with ans without endometriosis. The experiments seem well performed and the results are clear, and therefore merit publication.

However, in its present form the manuscript is difficult to read and to understand. Please consider the comments below a suggestions for improvement

  • A major problem is the clinical delineation of endometriosis, adhesions and possibly of the control group. Since deep endometriosis is predominantly classified as AFS II and III, while being a more severe form of endometriosis data would probably gain in clarity if women with deep endometriosis were considered separately.  Second, adhesions and cystic ovarian endometriosis are strongly associated: the authors are suggested to evaluate whether their adhesion group is more than only cystic ovarian endometriosis.  Three is seems useful to discuss that women with infertility without endometriosis might be a group which is endometriosis prone ie with similar genetic or epigenetic defects as women who develop endometriosis

Minor suggestions

  • It is strongly sugested to remain descriptive eg l 14 and others: many factors are associated with endometriosis; it is unclear which contribute to the pathogenesis of endometriosis, and it is even less clear which favor the initiation and which are involved in the growth of endometriosis
  • It might be discussed whether CTLA-4  causes endometriosis to be self limiting with most of typical lesions presenting as burnth-out
  • It is unclear whether endometriosis is associated with high estrogens

Text : In order to be readable by clinicians, it is suggested to add some explanation in the text. Examples  

  • For a clinician notions as immune tolerance, and overload of the immune system are not immediately clear
  • L80-85 rather belongs to discussion
  • Fig 1 is not relevant. Important is to define better the groups  
  • L111 why cephalic vein
  • L113-118 why 3 tubes .  Quid blood contaminated peritoneal fluid
  • It is suggested to explain to the clinician that many M&M are standard procedures whereas others are specific
  • “cells within the CD4+ 128 T, CD8+ T, and CD19+ B lymphocyte populations.” Add some explanation. Idemfor “CD4+CD25+high fork-141 head box P3 (FOXP3+) Treg subpopulation. During analysis, the CD3+CD16+CD56+ nat-142 ural killer T-like (NKT-like) cells population and CD16+CD56+ NK+ cells were also meas-143 ured with anti-CD3-FITC, CD16CD56-PE, and CD45- peridinin-chlorophyll-protein 144 (PerCP) mAbs (BD Biosciences, USA).”
  • Fif 2 : cannot be understood by a clinician
  • Fig 3: I doubt whether useful for this article
  • L178 outliers ?
  • Table 2 correlation: pearson or Spearman
  • Fig 4 ?? maybe add the stage of endometriosis

In conclusion, this manuscript merits publication, but needs revision.

Author Response

REVIEWER 1

Dear Reviewer,

We would like to kindly thank you for your prompt and valuable review of our paper. We agree, that the revisions were needed to improve the quality of the paper, therefore please find beneath the detailed description of the revisions according to your kind review.

Thank you again, we are truly hoping that now the paper will fulfill your requirements.

------

The manuscript “CLTA-4 expression is associated with the maintenance of 2 chronic inflammation in endometriosis and infertility” describes the expression of CTLA-4 on T and B cells, as well as the level of this marker in blood serum and peritoneal fluid, in women with and without endometriosis. The experiments seem well performed and the results are clear, and therefore merit publication.

However, in its present form the manuscript is difficult to read and to understand. Please consider the comments below a suggestions for improvement

  • A major problem is the clinical delineation of endometriosis, adhesions and possibly of the control group. Since deep endometriosis is predominantly classified as AFS II and III, while being a more severe form of endometriosis data would probably gain in clarity if women with deep endometriosis were considered separately.  

RE: Thank you for this important comment. We agree that a subanalysis of patients with deep infiltrating endometriosis would be very meaningful. However, in our study, we had only 11 patients with deep infiltrating endometriosis. For such a low sample size, the analysis could be biased and inconclusive. The issue mentioned could be addressed in a following study, further expanding the role of CTLA-4 in diagnosis, differentiation and treatment of endometriosis.

  • Second, adhesions and cystic ovarian endometriosis are strongly associated: the authors are suggested to evaluate whether their adhesion group is more than only cystic ovarian endometriosis.  

RE: Thank you for raising this issue. We characterized our study group in more detail in the Results section:

Lines 197-199: According to the rASRM scale, 17 (31%) patients were classified as Stage I, 17 (31%) as Stage II, 9 (17%) as Stage III, and 11 (20%) as Stage IV.

Lines 202-203: Eleven patients had deep infiltrating endometriosis. Among 31 patients with peritoneal adhesions, 20 were diagnosed with endometrioma.

  • Three is seems useful to discuss that women with infertility without endometriosis might be a group which is endometriosis prone ie with similar genetic or epigenetic defects as women who develop endometriosis

RE: We are grateful to the Reviewer for raising this important issue. We included a paragraph concerning the role for CTLA-4 in the pathogenesis of endometriosis-related infertility.

Lines 423-433: Although the clinical relationship between endometriosis and infertility is evident, the casual link between endometriosis and infertility is still poorly established [33]. Anupa et. al in their recent study showed that secretory profile of inflammation-associated cytokines in eutopic endometrium derived from patients suffering from severe ovarian endometriosis during their ‘window of implantation” differs from other infertile patients [34]. CCL3, CCL4, CCL5, CXCL10, FGF2, IFNG, IL1RN, IL5, TNFA, and VEGF were detected exclusively in stromal cells isolated from patients with endometriosis, and therefore might provide useful tool for distinguishing the source of infertility [34]. Our findings highlight the potential implication of CTLA-4 in the pathogenesis of endometriosis-related infertility and expression of CTLA-4 antigen on lymphocytes as a putative biomarker of this condition.

  1. M. B. Evans and A. H. Decherney, "Fertility and Endometriosis," Clin. Obstet. Gynecol. 2017, vol. 60, no. 3, pp. 497-502. doi: 10.1097/GRF.0000000000000295.

  1. G. Anupa et al., “Endometrial stromal cell inflammatory phenotype during severe ovarian endometriosis as a cause of endometriosis-associated infertility,” Reprod. Biomed. Online 2020, vol. 41, no. 4, pp. 623-639. doi: 10.1016/j.rbmo.2020.05.008.

Minor suggestions

  • It is strongly sugested to remain descriptive eg l 14 and others: many factors are associated with endometriosis; it is unclear which contribute to the pathogenesis of endometriosis, and it is even less clear which favor the initiation and which are involved in the growth of endometriosis

RE: We rephrased the mentioned sentence and other fragments which could sound elusive.

Lines 14-16: A number of factors are distinguished in the pathogenesis of endometriosis, including a large role attributed to altered immune mechanisms. Altered immune mechanisms are implicated in the pathogenesis of endometriosis.

  • It might be discussed whether CTLA-4  causes endometriosis to be self limiting with most of typical lesions presenting as burnth-out

RE: Thank you for this interesting hypothesis. However, we think that raising this issue in the manuscript will be too speculative and not directly related to our findings.

  • It is unclear whether endometriosis is associated with high estrogens

RE: We clarified this issue throughout the manuscript.

 Lines 33-34: The growth of endometrial tissue is estrogen dependent closely associated with steroid metabolism; therefore, it is estimated that the disease affects approximately 10% of all women of reproductive age [1,2,3].

Lines 42-44: Although chronic inflammation and high estrogen levels estrogen dependency are well-established factors in the characteristics of endometriosis, the exact etiology of the disease remains largely elusive [13, 14].

  1. E. Chantalat et al., “Estrogen Receptors and Endometriosis,” Int. J. Mol. Sci. 2020, vol. 21, no. 8, 2815. https://doi.org/10.3390/ijms21082815

Text : In order to be readable by clinicians, it is suggested to add some explanation in the text. Examples  

  • For a clinician notions as immune tolerance, and overload of the immune system are not immediately clear

RE: Thank you for this comment. From our experience, both terms are consistently used in the present literature related to immunology. It would be difficult to find alternatives which could briefly describe their full meanings, so we decided to leave them in their current form in the manuscript (they appear only three times throughout the whole manuscript).

  • L80-85 rather belongs to discussion

RE: Thank you for this valuable remark. We transferred the whole paragraph to the Discussion section (lines 394-400).

  • Fig 1 is not relevant. Important is to define better the groups  

RE: We removed Figure 1 from the manuscript. We described our study group in more detail in the Results section.

Lines 197-199: According to the rASRM scale, 17 (31%) patients were classified as Stage I, 17 (31%) as Stage II, 9 (17%) as Stage III, and 11 (20%) as Stage IV.

Lines 202-203: Eleven patients had deep infiltrating endometriosis. Among 31 patients with peritoneal adhesions, 20 were diagnosed with endometrioma.

  • L111 why cephalic vein

RE: Thank you for pointing out this inconsistency. The blood samples were collected from the most accessible vein at cubital fossa. We clarified this issue in the Methods section.

Lines 114-116: The material for the research was peripheral blood (PB) taken from the cephalic vein or other superficial vein with the best access for phlebotomy at cubital fossa from the patients in the total amount of 15 ml.

  • L113-118 why 3 tubes .  Quid blood contaminated peritoneal fluid

RE: We specified the reason for the technique used for collection of peritoneal fluid in the Methods section.

Lines 121-122: The peritoneal fluid was collected immediately after insertion of the laparoscope to avoid blood contamination.

  • It is suggested to explain to the clinician that many M&M are standard procedures whereas others are specific

RE: Thank you for this suggestion. The following data, measurements and laboratory tests are standard procedures in our clinic:

- presence of endometrioma

- presence of DIE (deep infiltrating endometriosis)

- presence of peritoneal endometriosis

- assessment of the stage of endometriosis

- presence of adhesions

- presence of pelvic pain

- presence of infertility

- Ca-125 concentration

- HE4 concentration

- assessment of the complete blood count

  • “cells within the CD4+ 128 T, CD8+ T, and CD19+ B lymphocyte populations.” Add some explanation. Idemfor “CD4+CD25+high fork-141 head box P3 (FOXP3+) Treg subpopulation. During analysis, the CD3+CD16+CD56+ nat-142 ural killer T-like (NKT-like) cells population and CD16+CD56+ NK+ cells were also meas-143 ured with anti-CD3-FITC, CD16CD56-PE, and CD45- peridinin-chlorophyll-protein 144 (PerCP) mAbs (BD Biosciences, USA).”

RE: Thank you for this suggestion. We have corrected this sentences as follows: We used fluorochrome-conjugated mAbs against the following markers: CD45- fluorescein isothiocyanate (FITC)/CD14- phycoerythrin (PE), mouse anti-human CD3-CyChrome, mouse anti-human CD19-FITC, mouse anti-human CD4-FITC, mouse anti-human CD8-FITC, mouse anti-human CTLA-4-PE-Cyanine 5 (Cy5) (BD Biosciences, San Jose, CA, USA). We also used the Human Treg Flow kit (FOXP3 Alexa Fluor 488/CD4 PE-Cy5 Cy5/CD25 PE; BioLegend, San Diego, CA, USA) to identify the CD4+CD25+high forkhead box P3 (FOXP3+) Treg subpopulation. CD3+CD16+CD56+ natural killer T-like (NKT-like) cells population and CD16+CD56+ NK+ cells were also measured with anti-CD3-FITC, CD16CD56-PE, and CD45- peridinin-chlorophyll-protein (PerCP) mAbs (BD Biosciences, USA).

  • Fif 2 : cannot be understood by a clinician

RE: Thank you for pointing out to this issue. We provided additional information in the Figure legend which in our opinion will increase its overall comprehensibility.

Lines 157-160: Figure 21. Example of flow cytometry analysis - aAssessment of the percentage of CD4 + T cells, CD8 + T cells and CD19 + B cells with CTLA-4 expression in patients with endometriosis. PBMCs from endometriosis patients were isolated and labeled as described in materials and methods. Scatter dot plots represent data from PBMCs in one endometriosis patient, analyzed by flow cytometry for CTLA-4 expression.

  • Fig 3: I doubt whether useful for this article

RE: Thank you for this suggestion. We removed Figure 3 from the manuscript.

  • L178 outliers ?

RE: We clarified this sentence accordingly.

Lines 182-185: The relationships between the variables were assessed on the basis of the Pearson's r R correlation coefficient, and in the case of variables with outliers if outliers were detected - additionally using the Spearman's rank correlation.

  • Table 2 correlation: pearson or Spearman

RE: Thank you for this comment. In the revised version of the manuscript, we thoroughly rebuilt the whole Results section. Table 2 is now removed from the manuscript, and all necessary information are provided in the text. It was Spearman correlation used, and we highlighted this information in the relevant paragraph of the Results section.

Lines 228-239: Patients with diagnosed endometritis endometriosis were divided into three groups depending on the stage of rASRM. The first group 1 (Stage I) and group 2 (Stage II) were was represented by 17 people patients each, group 2 (Stage II) also by 17, while and the third group 3 (Stage III-IV) was qualified by 20 people individuals. The study showed aA positive correlation between the severity of EMS endometriosis and the percentage of CD4 + / CTLA-4 + T cells (Spearman’s R = 0.531, p < 0.001), and a positive correlation between the severity of EMS endometriosis and the percentage of CD8 + / CTLA-4 + T cells (Spearman’s R = 0.450, p = 0.001) were observed. Moreover, there was also a weak positive correlation between the The severity of the disease of EMS endometriosis was weakly correlated and with the percentage of CD19 + / CTLA-4 + B cells (Spearman’s R = 0.315, p = 0.02). All Spearman's correlation coefficients are presented in Table 2.

  • Fig 4 ?? maybe add the stage of endometriosis

RE: Such an analysis would be indeed very informative. However, if additional grouping by stage of endometriosis was performed in our study, the resulting subgroups would be too small to be conclusively analyzed. This issue represents the area of potential further research aimed at women with endometriosis and infertility.

Reviewer 2 Report

Major

  1. The article is written as a progress report. The authors spend much time describing the data, but there is no analytical logic and hypothesis for each step. Even did not make any comment or summary on each finding makes it hard to read. However, the style of the present results makes them confused and overstated. For example, in the abstract, "we confirmed that negative costimulation of this molecule in inducing the maintenance of chronic inflammation in endometriosis." The problem is the data link to negative costimulation and inducting the maintenance of chronic inflammation are unclear. It might "CTLA-4 high expression" equal to "negative costimulation"; and "expression correlates to disease stages" equal to "inducting the maintenance of chronic inflammation." The sample marker analysis is not equal to the biological response.
  2. Most discussions are repeated from the introduction. The author did not discuss the study value and new findings compare to other studies. Three references that described the same findings are not included in the article. PMID: 32593225, which discussed the sCTLA-4 in local immune checkpoint and endometriosis; PMID: 32002820, which showed women endometriosis had higher percentages of TNFRII+ Treg and CTLA-4+ Treg in their peripheral blood. Hou XX, Wang XQ, Li DJ. Roles of regulatory T Cells in pathogenesis of endometriosis. Reprod Dev Med [serial online] 2019 [cited 2021 Jan 28];3:117-23. Available from: https://www.repdevmed.org/text.asp?2019/3/2/117/262392. Authors need to highlight the novelty and importance of research by themselves.
  3. Table 2. is analyzed the correlation between the severity of EMS and CTLA-4 lymphocytes. The 20 patients in the control group should be treated as stage 0 and put into the model.
  4. The logistic regression model is suggested to analyze the set of data.

Minor

  1. The data presentation of correlation analysis (Table 2 vs. Figure 4-6) is not consistent. If it is plotted, the R- and P-value are suggested to annotate to read clearly in each plot.
  2. The legends of Figure 4-6 is too simple. At least, the analysis population should be illustrated.
  3. The article is written as a progress report. Many sentences are repeated from copy and paste. For example, the subtitle of 3.2.2 and 3.2.3 is the same. The author forgot to change 3.2.3 by "pelvic pain syndrome" after pasting a paragraph. However, reading the same sentence is quite annoying.
  4. Due to the lack of peritoneal fluid of control in table 5, the result's meaning is unclear.
  5. The style of data presentation is not good. It can be simplified into 2 or 3 tables without taking up space to make it is easier to read.

Author Response

Dear Reviewer,

We would like to kindly thank you for your time and knowledge that you have invested in reading and revising our paper. We do agree, that the corrections were needed to improve the quality of the paper. We are now hoping, that you will accept the manuscript in its present form.

Please find beneath the detailed point-by-point answers to your concernes.

Thank you again,

Regards,

_______

Major

  1. The article is written as a progress report. The authors spend much time describing the data, but there is no analytical logic and hypothesis for each step. Even did not make any comment or summary on each finding makes it hard to read. However, the style of the present results makes them confused and overstated. For example, in the abstract, "we confirmed that negative costimulation of this molecule in inducing the maintenance of chronic inflammation in endometriosis." The problem is the data link to negative costimulation and inducting the maintenance of chronic inflammation are unclear. It might "CTLA-4 high expression" equal to "negative costimulation"; and "expression correlates to disease stages" equal to "inducting the maintenance of chronic inflammation." The sample marker analysis is not equal to the biological response.

RE: Thank you for this important comment. To address the issues raised, we thoroughly rebuilt the whole Results section. We removed 2 figures and 3 tables from the manuscript, leaving only those more relevant for our hypothesis. We corrected the abstract accordingly.

Lines 18-27: In this study, we examined the expression of CTLA-4 on T and B cells by flow cytometry, as well as and its levels the level of this marker in blood serum and peritoneal fluid, by flow cytometry and by ELISA. As a result, we showed statistically significant changes in the lLevels of CTLA-4+ T cells were significantly higher in patients with more advanced endometriosis than in those with less advanced disease, thanks to which we confirmed that negative costimulation of this molecule in inducing the maintenance of chronic inflammation in endometriosis. Additionally, the negative correlation of CTLA-4+ T lymphocytes and the percentage of NK and NKT-like cells in women with endometriosis and infertility may indicate a different etiopathogenesis of endometriosis accompanying infertility. Our researchOur findings sheds light on the potential of CTLA-4 to develop in developing new diagnostic therapeutic and therapeutic research approaches in endometriosis management.

  1. Most discussions are repeated from the introduction. The author did not discuss the study value and new findings compare to other studies. Three references that described the same findings are not included in the article. PMID: 32593225, which discussed the sCTLA-4 in local immune checkpoint and endometriosis; PMID: 32002820, which showed women endometriosis had higher percentages of TNFRII+ Treg and CTLA-4+ Treg in their peripheral blood. Hou XX, Wang XQ, Li DJ. Roles of regulatory T Cells in pathogenesis of endometriosis. Reprod Dev Med [serial online] 2019 [cited 2021 Jan 28];3:117-23. Available from: https://www.repdevmed.org/text.asp?2019/3/2/117/262392. Authors need to highlight the novelty and importance of research by themselves.

RE: Thank you for pointing out to these interesting papers. In response, we introduced major modifications in the Discussion section, adding four recent references on the topic. We included additional sections describing novel findings on CTLA-4 expression on Tregs (a) and soluble CTLA-4 (b) in endometriosis. Additionally, we expanded one paragraph focusing on co-existence of endometriosis and infertility (c).

  1. (a)

Lines 444-451: Interestingly, Santoso et al. revealed significantly higher peritoneal concentration of soluble CTLA-4 in infertile endometriosis patients than in infertile patients without endometriosis [38]. The differences in serum CLTA-4 levels were reported only be-tween late-stage endometriosis and non-endometriosis patients [38]. Our own studiesstudy did not showed any significant changes differences in plasma CTLA-4its concentration between patients with endometriosis and the control group, but no relationship between endometriosis stage and levels of CTLA-4 both in the peripheral blood and in the peritoneal fluid were observed.

  1. B. Santoso et al., “Soluble immune checkpoints CTLA-4, HLA-G, PD-1, and PD-L1 are associated with endometriosis-related infertility,” Am. J. Reprod. Immunol. 2020, vol. 84, no. 4, e13296. doi: 10.1111/aji.13296.
  2. (b)

Lines 413-416: There are no studies on CTLA-4 expression in endometriosis. Recently, patients with endometriosis were reported to have higher percentage of Treg cells expressing CTLA-4 in plasma than healthy individuals [32]. No such relationship was found in Treg cells isolated from peritoneal fluid [32].

  1. B. Y. Gueuvoghlanian-Silva, C. Hernandes , R. P. Correia, and S. Podgaec, “Deep Infiltrating Endometriosis and Activation and Memory Surface Markers and Cytokine Expression in Isolated Treg Cells,” Reprod. Sci. 2020, vol. 27, no. 2, pp. 599-610. doi: 10.1007/s43032-019-00060-1.
  2. (c)

Lines 423-433: Although the clinical relationship between endometriosis and infertility is evident, the casual link between endometriosis and infertility is still poorly established [33]. Anupa et. al in their recent study showed that secretory profile of inflammation-associated cytokines in eutopic endometrium derived from patients suffering from severe ovarian endometriosis during their ‘window of implantation” differs from other infertile patients [34]. CCL3, CCL4, CCL5, CXCL10, FGF2, IFNG, IL1RN, IL5, TNFA, and VEGF were detected exclusively in stromal cells isolated from patients with endometriosis, and therefore might provide useful tool for distinguishing the source of infertility [34]. Our findings highlight the potential implication of CTLA-4 in the pathogenesis of endometriosis-related infertility and expression of CTLA-4 antigen on lymphocytes as a putative biomarker of this condition.

  1. M. B. Evans and A. H. Decherney, "Fertility and Endometriosis," Clin. Obstet. Gynecol. 2017, vol. 60, no. 3, pp. 497-502. doi: 10.1097/GRF.0000000000000295.

  1. G. Anupa et al., “Endometrial stromal cell inflammatory phenotype during severe ovarian endometriosis as a cause of endometriosis-associated infertility,” Reprod. Biomed. Online 2020, vol. 41, no. 4, pp. 623-639. doi: 10.1016/j.rbmo.2020.05.008.
  1. Table 2. is analyzed the correlation between the severity of EMS and CTLA-4 lymphocytes. The 20 patients in the control group should be treated as stage 0 and put into the model.

RE: We would tend to disagree with proposed approach. The analysis mentioned aimed at  determination of association between disease severity and levels of CTLA-4 expression rather than determination of association of CTLA-4 with endometriosis, which was done in the previous step of statistical analysis. We believe that the applied approach better reflects the aim of this particular analysis.

  1. The logistic regression model is suggested to analyze the set of data.

RE: Ordered logistic regression which would need to be applied (non-binary outcome in this case, 1-3 ordered categories) applies maximum likelihood estimates to calculate the coefficients rather than the ordinary least square method. Therefore, it demands relative large sample sizes. In our study, with number of observations in subgroups ranging from 17 to 20, this issue could have biased the analysis. In our opinion, Spearman’s correlation applied might be a better suited option for this particular analysis.

Minor

  1. The data presentation of correlation analysis (Table 2 vs. Figure 4-6) is not consistent. If it is plotted, the R- and P-value are suggested to annotate to read clearly in each plot.

RE: Thank you for raising this issue. In the revised version of the manuscript, the R and p values are shown on all plots presenting correlation.

Lines 264-268: A positive correlation of the percentage of CD4+CTLA-4+ T cells with the number of monocytes was observed (r = 0.564, p = 0.005) (Figure 4a2a). However, in the case of CD4+CTLA-4+ T cells with the percentage of CD3-CD16+CD56+ NK cells (r = −0.669, p <0.001) (Figure 4b2b), and the percentage of NKT-like CD3 + CD16 + CD56 + cells (r = - 0.454, p = 0.029), negative correlations were shown (Figure 4c2c).

Lines 274-280: A positive correlation between the percentage of CD8+CTLA-4+ T cells and the number of monocytes was observed (r = 0.465, p = 0.025) (Figure 5a3a). There was a negative correlation of CD8+CTLA-4+ T cells with the percentage of CD3-CD16+CD56+ NK cells (r = −0.485, p = 0.019) (Figure 5b3b) and the percentage of NKT-like CD3+CD16+CD56+ cells (r = −0.462, p = 0.027 ) (Figure 5c3c). Moreover, a positive correlation was also found between the percentage of CD8+CTLA-4+ T cells and the percentage of CD4+CD25+ highFoxp3 regulatory T cells (r = 0.522, p = 0.011) (Figure 5d3d).

Lines 287-289: A negative correlation between CD19+CTLA-4+ B lymphocytes and the percent-age of CD3-CD16+CD56+ NK cells was found to demonstrate a negative correlation (r = −0.521, p = 0.011) (Figure 64).

  1. The legends of Figure 4-6 is too simple. At least, the analysis population should be illustrated.

RE: Thank you for raising this issue. We included more details in the figure legends.

Lines 271-273: Figure 42. Correlation between the percentage of CD4+ T cells expressing CTLA-4 (CD4+CTLA-4+)T cells and: (a) and the number of monocytes; (b) and the percentage of NK cells (CD3-CD16+CD56+); (c) and the percentage of NKT-like cells (CD3+CD16+CD56+) cells. in patients with endometriosis and coexisting infertility (n = 27)

Lines 283-286: Figure 53. Correlation between the percentage of CD8+ T cells expressing CTLA-4 (CD8+CTLA-4+) T cells and: (a) and the number of monocytes; (b) and the percentage of NK cells (CD3-CD16+CD56+); (c) and the percentage of NKT-like (CD3+CD16+CD56+) cells; (d) and the percentage of subpopulation of regulatory T cells with high forkhead box P3 expression (CD4+CD25+highFoxp3+). in patients with endometriosis and coexisting infertility (n = 27)

Lines 292-293: Figure 64. Correlation between the percentage of B lymphocytes expressing CTLA-4 (CD19+CTLA-4+) B lymphocytes and the percentage of NK cells (CD3-CD16+CD56+) NK cells. in patients with endometriosis and coexisting infertility (n = 27)

  1. The article is written as a progress report. Many sentences are repeated from copy and paste. For example, the subtitle of 3.2.2 and 3.2.3 is the same. The author forgot to change 3.2.3 by "pelvic pain syndrome" after pasting a paragraph. However, reading the same sentence is quite annoying.

RE: Thank you for this important comment. We thoroughly re-written the whole Results section to address it. We renamed the subtitles and improved the wording to remove repetitions. Due to space constraints, here we present only the examples of modifications introduced.

Lines 294-301:

3.2.2. Assessment of the relationship between the pPercentage of T and B lymphocytes expressing the CTLA-4 molecule and selected parameters of the specific and non-specific response in patients with endometriosis and accompanying adhesive disease.

In patients with endometriosis and coexisting adhesive disease, nNegative correlations of T lymphocytes CD4+CTLA-4+ with the percentage of NK cells CD3-CD16+CD56+ were was demonstrated (r R = -0.622, p < 0.001). A correlation was also observed between the The percentage of T cells CD4+CTLA-4+ and was negatively correlated with the percentage of regulatory T cells CD4+CD25+highFoxp3+ (r R = -0.45, p = 0.013).

Lines 307-309:

3.2.3. Assessment of the relationship between the pPercentage of T and B lymphocytes expressing the CTLA-4 molecule and selected parameters of the specific and non-specific response in patients with endometriosis and accompanying adhesive disease pelvic pain syndrome.

  1. Due to the lack of peritoneal fluid of control in table 5, the result's meaning is unclear.

RE: We totally agree with that the lack of the control peritoneal fluid is a limitation of this analysis. However, surgical collection of peritoneal fluid during laparoscopy is primary done in individuals diagnosed with some kind of abnormalities, e.g., inflammation or cancer. Our control group consisted of healthy individuals, so peritoneal fluid samples were not available. This limitation could be addressed in an another study focused on comparison of soluble CTLA-4 levels in peritoneal fluid in patients with different types of gynecologic disorders with indication of laparoscopy.

  1. The style of data presentation is not good. It can be simplified into 2 or 3 tables without taking up space to make it is easier to read.

RE: Thank you for raising this issue. We thoroughly reorganized and re-written the whole Results section. We removed one figure and three tables, incorporating the relevant information in the text or on the Figures. In the revised version of the manuscript, three figures and two tables are included in the Results section. For space constraints, we do not show all the changes made here. All modifications can be followed as track changes option in the revised manuscript.

Round 2

Reviewer 2 Report

none